# CRL2$^{ZER1/ZYG11B}$ recognizes small N-terminal residues for degradation

Yao Li[1,5], Yueling Zhao[2,5], Xiaojie Yan ®[1,5], Chen Ye[2,5], Sara Weirich[3], Bing Zhang[1], Xiaolu Wang[4], Lili Song[2], Chenhao Jiang[2], Albert Jeltsch[3], Cheng Dong ®[1] ✉ & Wenyi Mi ®[2] ✉

N-degron pathway plays an important role in the protein quality control and maintenance of cellular protein homeostasis. ZER1 and ZYG11B, the substrate receptors of the Cullin 2-RING E3 ubiquitin ligase (CRL2), recognize N-terminal (Nt) glycine degrons and participate in the Nt-myristoylation quality control through the Gly/N-degron pathway. Here we show that ZER1 and ZYG11B can also recognize small Nt-residues other than glycine. Specifically, ZER1 binds better to Nt-Ser, -Ala, -Thr and -Cys than to -Gly, while ZYG11B prefers Nt-Gly but also has the capacity to recognize Nt-Ser, -Ala and -Cys in vitro. We found that Nt-Ser, -Ala and -Cys undergo Nt-acetylation catalyzed by Nt-acetyltransferase (NAT), thereby shielding them from recognition by ZER1/ZYG11B in cells. Instead, ZER1/ZYG11B readily targets a selection of small Nt-residues lacking Nt-acetylation for degradation in NAT-deficient cells, implicating its role in the Nt-acetylation quality control. Furthermore, we present the crystal structures of ZER1 and ZYG11B bound to various small Nt-residues and uncover the molecular mechanism of non-acetylated substrate recognition by ZER1 and ZYG11B.

N-terminal acetylation (Nt-Ac) is a prevalent protein modification that occurs on approximately 90% of human proteins[1,2]. It is catalyzed by Nt-acetyltransferases (NATs), which transfer the acetyl group from acetyl coenzyme A onto the free α-amino group of protein Nt-residues. This reaction mainly takes place co-translationally on nascent polypeptide chains, but it can also occur post-translationally for example on actin[3,4]. In contrast to acetylation of ε-amino groups on internal lysine residues, Nt-Ac is considered irreversible due to the lack of an Nt-deacetyltransferase enzyme in human cells[5].

The NAT family currently encompasses eight eukaryotic members (NatA to NatH) with distinctive features, of which NatA is the major NAT that accounts for the Nt-Ac of ~40% of the human proteome[4]. The NatA complex, which consists of the catalytic subunit NAA10 and the auxiliary subunit NAA15, catalyzes acetylation of N-termini starting with a small residue (Ser, Ala, Thr, Cys, and infrequently Val and Gly), which are exposed after the removal of initiator methionine (iMet) by methionine aminopeptidases (MetAPs)[6].

The resulting Nt-Ac changes the chemical properties of the N-terminus by neutralizing its positive charge, and generally prevents other modifications of the N-terminus[7]. Therefore, it is not surprising that Nt-Ac affects a wide range of protein functions, including protein stability, folding, protein-protein interactions and subcellular localization[4]. In particular, the Nt-Ac acts as a degradation signal (degron) that is targeted by E3 ubiquitin ligase (Ac/N-recognin) Doa10

[1]The Province and Ministry Co-sponsored Collaborative Innovation Center for Medical Epigenetics, Key Laboratory of Immune Microenvironment and Disease (Ministry of Education), Tianjin Medical University General Hospital, The Second Hospital of Tianjin Medical University, Department of Biochemistry and Molecular Biology, Tianjin Medical University, Tianjin 300070, China. [2]The Province and Ministry Co-sponsored Collaborative Innovation Center for Medical Epigenetics, Key Laboratory of Immune Microenvironment and Disease (Ministry of Education), Tianjin Medical University General Hospital, Department of Immunology, Tianjin Medical University, Tianjin 300070, China. [3]Department of Biochemistry, Institute of Biochemistry and Technical Biochemistry, University of Stuttgart, Allmandring 31, 70569 Stuttgart, Germany. [4]Department of Pharmacology, School of Basic Medical Sciences, Tianjin Medical University, Tianjin 300070, China. [5]These authors contributed equally: Yao Li, Yueling Zhao, Xiaojie Yan, Chen Ye. ✉e-mail: dongcheng@tmu.edu.cn; wenyi.mi@tmu.edu.cn

or Not4 for proteasomal degradation through the Ac/N-degron pathway (formerly "Ac/N-end rule pathway")[8,9]. Proteins containing Ac/N-degrons with Nt-acetylated Met, Ala, Val, Ser, Thr, or Cys residue tend to be conditionally active in the physiologically relevant manner since they may be sterically shielded by interaction partners[8,9]. The Ac/N-degron pathway therefore plays important roles in the control of protein quality and stoichiometry[9,10].

At the same time, the Nt-acetyl group is thought to confer stability to some proteins[11], such as the Drosophila Hyx protein[12], the human THOC7 protein[13] and a fraction of yeast proteins[11,14]. A recent study has shown that certain substrates of NatA can be recognized by IAP (inhibitor of apoptosis protein) ubiquitin ligase in vitro, implicating that Nt-Ac prevents protein degradation in human cells[15]. Another study found that the co-translational Nt-Ac promotes proteome stability in plants, suggesting that a non-Ac/N-degron exists in plants as well albeit the respective E3 ligase yet to be identified[16].

Nt-Gly is a potential substrate of NatA, but it is frequently Nt-myristoylated by N-myristoyltransferase (NMT1/2) instead of being Nt-acetylated[1,4,17]. The Nt-myristoylated glycine mediates the proper membrane association and localization of target proteins[18]. Proteins with Nt-glycine that failed to be myristoylated can be recognized by ZYG11B and ZER1, the substrate receptors of Cullin 2-RING E3 ubiquitin ligase, and targeted for degradation through the Gly/N-degron pathway[19]. Although ZYG11B and ZER1 specifically recognize the free Nt-glycine, GPS (global protein stability) assay indicated that ZER1 might recognize substrates starting with residues other than glycine[19], implying a complex the substrate recognition of ZER1.

In this study, we comprehensively investigated the substrate preferences of ZER1 and ZYG11B, and found that they have the ability to recognize small Nt-residues, an overlap of NatA substrates, other than glycine in vitro. Remarkably, ZER1 revealed a stronger binding to Nt-Ser, -Ala, -Thr and -Cys compared to -Gly. We provide structural insights into the non-glycine substrate recognition by ZER1 and ZYG11B. In addition, ZER1/ZYG11B is able to target the Nt-Ser, −Ala and -Cys degron for degradation upon NAT depletion in cells, suggesting that Nt-Ac by NAT shield target proteins from recognition by ZER1/ZYG11B in order to maintain protein homeostasis.

## Results

### ZER1 and ZYG11B can recognize small Nt-residues besides glycine

ZER1 and ZYG11B were identified as the Gly/N-recognins that are responsible for recognition of the Gly/N-degron[19,20]. Although ZER1 and ZYG11B share the majority of the substrates and collaborate in the degradation of proteins starting with Nt-Gly, ZYG11B likely bears more responsibility and selectivity for the recognition of Nt-Gly, and ZER1 might, in some contexts, recognize substrates that begin with residues other than glycine[19]. Indeed, upon inspection of the crystal structures of ZER1 bound to Gly/N-degrons, we found that the Nt-Gly is buried at the bottom of the cavity lined with multiple water molecules[20], raising the possibility that the Gly-binding space could be expanded if there were rearrangements of water molecules.

Therefore, we speculated that certain small residues besides glycine might be tolerated in this cavity of ZER1. To address this conjecture, we performed a mutation-scanning peptide SPOT array to explore the substrate specificity of ZER1 as well as ZYG11B. In this experiment, a 15-mer peptide (sequence: GFLHVGQDGLELPTS) derived from the N-terminus of ZNF701 was systematically substituted at the first 8 positions with each of the 18 natural amino acids (Cys and Trp were excluded due to the difficulties in synthesis). The array was incubated with GST-fused ZER1 (residues 469–766) or ZYG11B (residues 490–744), and binding to the peptide spots was detected by chemiluminescence using anti-GST antibody (Fig. 1a, b).

Surprisingly, we found that ZER1 preferentially binds to Nt-Ser, -Thr and -Ala over -Gly, while ZYG11B displays robust binding to Nt-Gly,

-Ser and -Ala, with a slight selectivity toward -Gly. Moreover, ZER1 prefers bulky aromatic residues at both position 2 and position 3, whereas ZYG11B revealed a preference for aromatic residues at position 2 with minor dependence at the following positions (Fig. 1a–d, and Supplementary Fig. 1a, b). These observations are in agreement with a previous report[20].

To further investigate the binding specificity of ZER1 and ZYG11B, we synthesized a set of eight-residue peptides (sequence: XFLHVGQD, X = any free Nt-residue), and carried out isothermal titration calorimetry (ITC) assays to quantify their binding affinities for ZER1 and ZYG11B. In line with our SPOT-array results, ZER1 binds to XFLHVGQD peptides in which X is either Ser ($K_D$ of 1.3), Ala ($K_D$ of 1.6), Thr ($K_D$ of 3.1) or Cys ($K_D$ of 3.9) more potently than to the Gly-starting peptide ($K_D$ of 4.2) (Fig. 1e and Supplementary Fig. 1c). Meanwhile, ZYG11B displayed appreciable binding to Nt-Ser ($K_D$ of 6.8), -Ala ($K_D$ of 7.1), or -Cys ($K_D$ of 3.0), albeit slightly weaker than Nt-Gly ($K_D$ of 2.5) (Fig. 1e, and Supplementary Fig. 1e). Intriguingly, unlike ZER1, ZYG11B was unable to bind to Nt-Thr (Fig. 1e, and Supplementary Fig. 1c, e), although they adopt similar structural architectures[20]. The rest of Nt-residues were disallowed for the binding to both ZER1 and ZYG11B (Fig. 1e, and Supplementary Fig. 1c–f). In summary, our results demonstrate that ZER1 and ZYG11B can recognize a fraction of small Nt-residues, in which ZER1 prefers Nt-Ser, -Ala, -Thr and -Cys more than the canonical -Gly, while ZYG11B has a specificity toward Nt-Gly but it can also recognize -Ser, -Ala or -Cys in vitro.

### NatA-mediated Nt-Ac shields the substrates from recognition by ZER1/ZYG11B

To determine whether the small non-glycine residues (here we first focus on Nt-Ser and -Ala because of their strong binding affinities) can be N-degrons that are recognized by ZER1 and then targeted for protein degradation in cells, we constructed the Ub-GPS reporters[19] in which the 12-mer ZNF701 peptide and its otherwise identical Nt-Ser and -Ala substitutions were fused to the N-terminus of GFP (Fig. 2a). However, the results showed that, in contrast to Gly-GFP fusion protein, neither Ser-GFP fusion nor Ala-GFP fusion was destabilized upon overexpression of exogenous ZER1 in HEK293T cells (Fig. 2b). This is consistent with previous stability profiling of the human N-terminome analysis that the Nt-Gly but not the -Ser or -Ala can act as a potent Nt-degron[19], while inconsistent with the in vitro studies present above.

These distinctive observations led us to wonder what might be the reason behind such difference between findings in vitro and observations in cells. We have noticed that these ZER1-binding small Nt-residues are known to be the typical substrates of NatA. Particularly, the Nt-Ser and -Ala are frequently (99% and 95%) Nt-acetylated by NatA after iMet excision, whereas Nt-Gly is rarely Nt-acetylated in the human proteome[4,7]. Therefore, we suspected that in the physiological relevant context, the Nt-Ac modification of Ser or Ala prevents their recognition by ZER1. In other words, NatA may shield the proteins bearing Nt-Ser and -Ala from ZER1-mediated degradation through Nt-Ac modification.

To ascertain this, we first performed ITC assays to examine the interactions between ZER1 and ZYG11B with the otherwise identical but Nt-acetylated peptides. The results clearly showed that Nt-Ac modification caused complete loss of ZER1 and ZYG11B binding (Supplementary Fig. 2a, b). Next, we evaluated the cellular acetylation state of the Ser-GFP fusion by liquid chromatography mass spectrometry (LC-MS). Indeed, we found that 85.7% of Ser-GFP were Nt-acetylated under our experimental conditions (Supplementary Fig. 2c). These results help to explain why it is hard to detect Ser-GFP degradation upon ZER1 overexpression in cells.

To further extend this notion, we knocked down (KD) NAA10, the catalytic subunit of NatA, and generated stable KD cells (Supplementary Fig. 2d). Although the Nt-Ac levels of Ser-GFP remained constantly high even in NAA10-KD cells, the levels of non-acetylated proteins modestly increased compared to that of WT cells (Supplementary

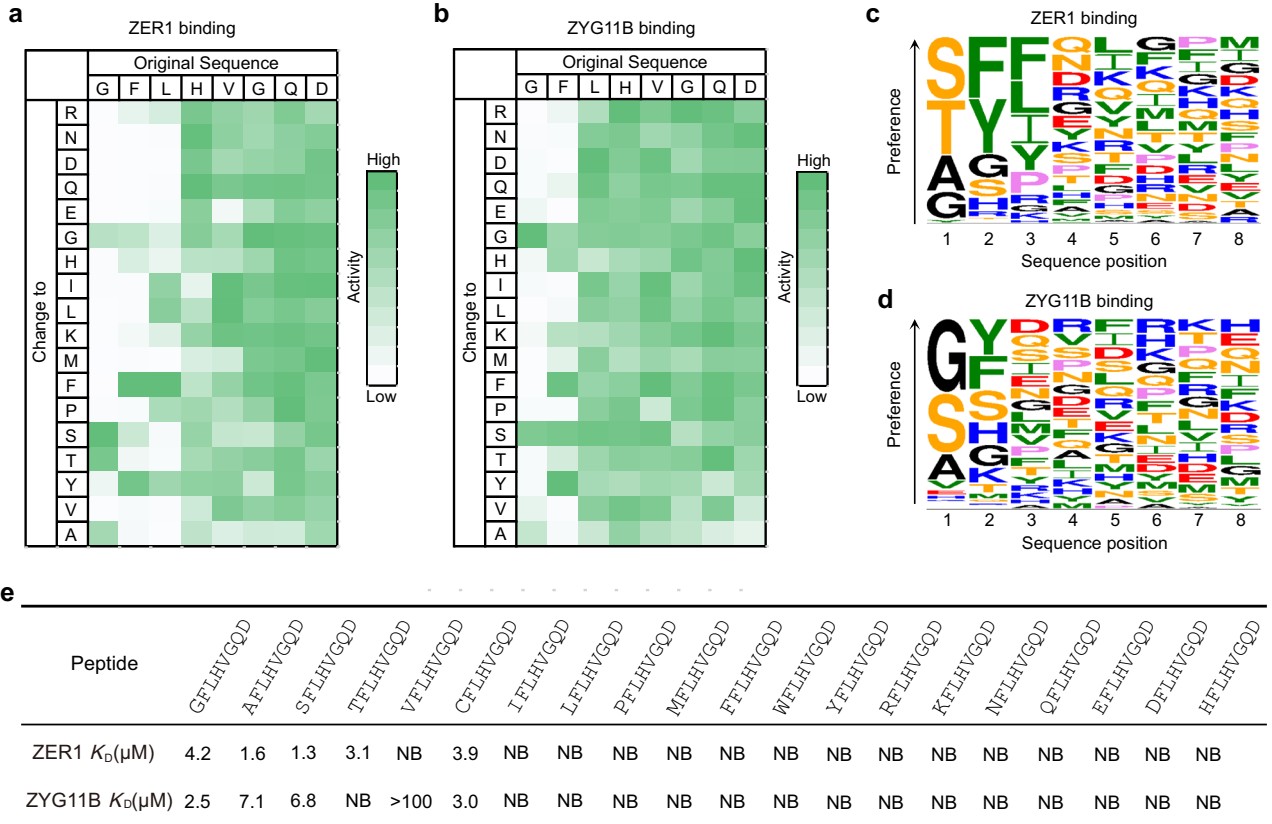

**Fig. 1 | ZER1 and ZYG11B can recognize non-glycine small Nt-residues.**
**a**, **b** Binding patterns of GST-tagged ZER1$_{469-766}$ and ZYG11B$_{490-744}$ to peptide SPOT arrays containing the GFLHVGQDGLELPTS peptide and its indicated derivatives containing mutations in the first 8 positions. The array binding data were quantified and the relative binding of ZER1 and ZYG11B to each peptide is shown, with strong to weak binding scaled from green to white. **c**, **d** Sequence logos of the substrate preference of ZER1 and ZYG11B. **e** Binding affinities ($K_D$s) of XFLHVGQD peptides (X is a set of 20 amino acids in the genetic code) peptides for the ZER1 (residues 469–766) or ZYG11B (residues 490–728). NB, no detectable binding. See also Supplementary Fig. 1.

Fig. 2c). Consistent with this, our co-immunoprecipitation (co-IP) assays showed that the amount of ZER1 binding with Ser-GFP or Ala-GFP increased obviously in NAA10-KD cells relative to the WT (Supplementary Fig. 2e). Remarkably, our GPS assay showed that the Ser-GFP and Ala-GFP fusion proteins were visibly degraded upon ZER1 overexpression in NAA10-KD cells in contrast to WT cells (Fig. 2b, c). Consistently, NAA10 KD significantly reduced the GFP/RFP ratio and promoted the degradation of both Ser- and Ala-GFP fusions in the GPS assay (Supplementary Fig. 2f).

Besides NatA, the Nt-Ser can also be Nt-acetylated by NatD which is highly selective for histones H2A and H4[21,22]. Our ITC data showed that the free H2A peptide (sequence: SGRGKQGGKA) can readily bind to ZER1 (Supplementary Fig. 2g). To explore whether the Nt-Ser motif derived from H2A can become a degron, we generated NatD-deficient cells via KD of the catalytic NAA40 subunit (Supplementary Fig. 2d). As expected, the Nt-Ser motif of H2A fused to GFP conferred instability upon the overexpression of ZER1 in NatD-deficient cells (Fig. 2d), further supporting that the non-Ac Nt-Ser is the potential N-degron recognized by ZER1.

Notably, ZER1 and ZYG11B also have the capacity to recognize the Nt-Cys, which is known to be only 50% Nt-acetylated by NatA in cells[7]. To test whether ZER1 and ZYG11B can directly target Nt-Cys for degradation in cells, we exogenously overexpressed ZER1 or ZYG11B in the Nt-Cys-GFP reporter cell lines. GPS assays showed that in contrast to control GFP, the Cys-GFP fusion proteins are indeed efficiently degraded upon the overexpression of ZER1 or ZYG11B (Fig. 2e). Moreover, the Cys-GFP exhibited dramatic instability in response to NAA10 KD, whereas further double knock out of ZER1/ZYG11B was

sufficient to restore Cys-GFP stability (Fig. 2f), establishing that the free Nt-Cys can be targeted by ZER1/ZYG11B for degradation.

Taken together, these results suggest that some small Nt-residues such as -Ser, -Ala and -Cys can conditionally create non-Ac/N-degrons, which are targeted by ZER1/ZYG11B for degradation. Because Ser- or Ala-starting proteins are almost always Nt-acetylated by NatA (with the exception of H2A and H4 which are specifically acetylated by NatD), the Nt-Ac modification results in preventing their recognition by ZER1/ZYG11B. Thus, we reason that ZER1/ZYG11B might act as an N-recognin and participate in "Nt-acetylation quality control" apart from "Nt-myristoylation quality control", by degrading proteins that bear small Nt-residue degrons conditionally exposed after failure of their Nt-Ac modification.

## Structural insight non-glycine Nt-residues recognition by ZER1/ZYG11B

To understand the molecular basis of ZER1/ZYG11B-mediated binding of non-Gly Nt-residue peptides, we determined the crystal structures of ZER1 bound to either SFLH-, AFLH- or TFLH-peptide (Supplementary Fig. 3a–c), as well as the crystal structures of ZYG11B bound to either SFLH-, AFLH- or CFLH- peptide (Supplementary Fig. 3d–f). Data collection and refinement statistics for all structures are summarized in Tables 1 and 2. Structural comparisons indicated that the peptide-binding modes of ZER1 and ZYG11B are similar (Supplementary Fig. 3). All the bound peptides are engaged in the common binding pocket formed by the ARM (armadillo) repeats[20].

In this cavity, the main chains of the first three residues of the bound peptides are coordinated by a set of conserved hydrogen bonds

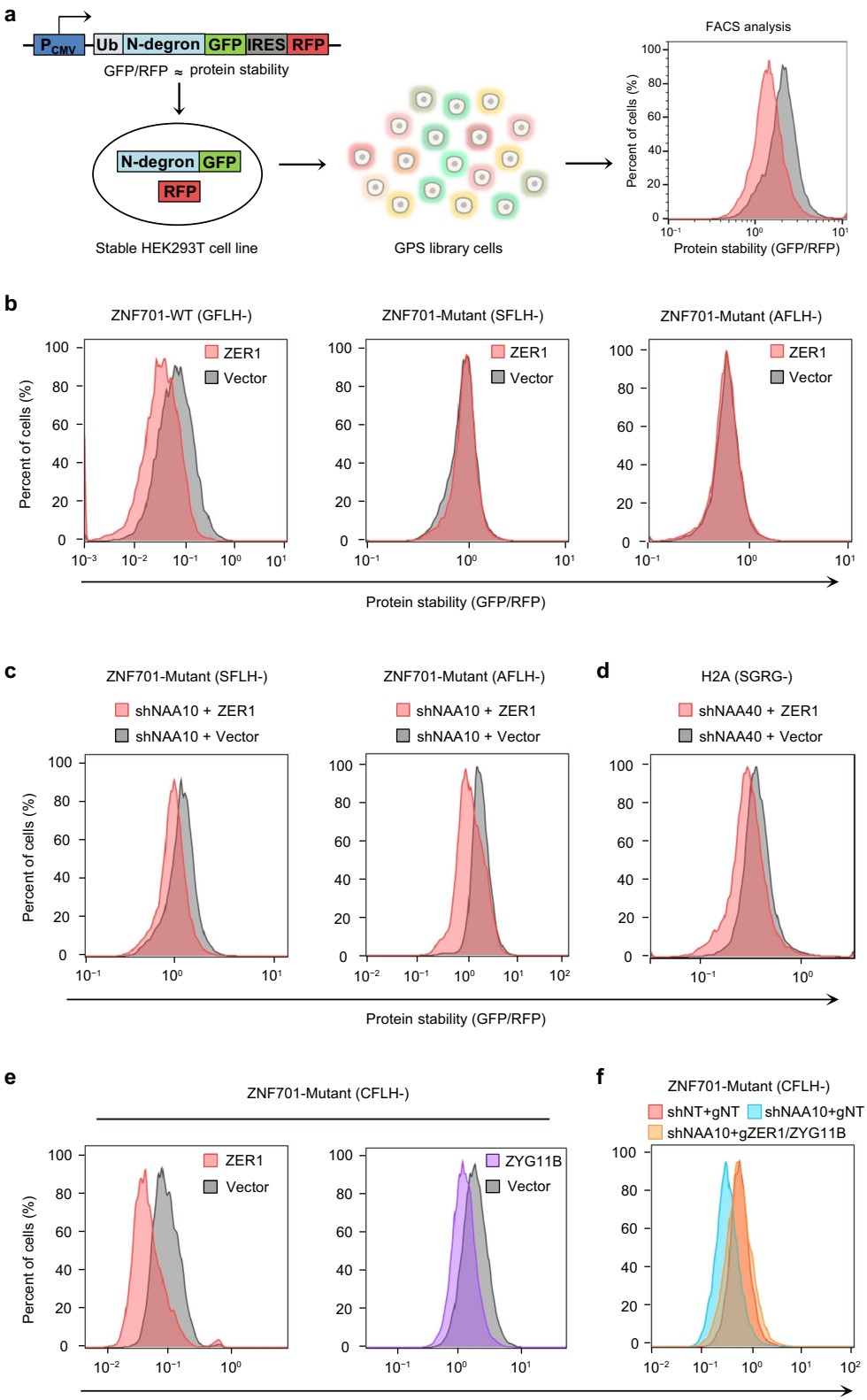

**Fig. 2 | Nt-acetylation of Ser, Ala or Cys prevents them from recognition by ZER1 or ZYG11B. a** Schematic of the global protein stability (GPS) assay. P$_{CMV}$, cytomegalovirus promoter, Ub ubiquitin, IRES internal ribosome entry site, GFP green fluorescent protein, RFP red fluorescent protein. **b** Stability analysis of GFLH-, SFLH- or AFLH-GFP upon ZER1 overexpression in HEK293T cells by GPS. The ratio of GFP/RFP was analyzed by flow cytometry. **c** Stability analysis of SFLH- or AFLH-fused GFP with full-length ZER1 overexpression in NAA10 knockdown cell lines. The ratio of GFP/RFP was analyzed by flow cytometry. **d** Stability analysis of N-terminal peptide derived from H2A with full-length ZER1 overexpression in NAA40 knock-down cell lines. The ratio of GFP/RFP was analyzed by flow cytometry. **e** Stability analysis of CFLH-fused GFP upon ZER1 or ZYG11B overexpression in HEK293T cells by GPS. The ratio of GFP/RFP was analyzed by flow cytometry. **f** Stability analysis of CFLH-fused GFP upon NAA10 knockdown or with simultaneous knockout of ZER1/ZYG11B in HEK293T cells. All FACS sequential gating images are provided in Supplementary Fig. 6.

**Table 1 | Data collection and refinement statistics of ZER1**

|  | ZER1-SFLH | ZER1-AFLH | ZER1-TFLH |
|---|---|---|---|
| PDB accession code | 7XYS | 7XYT | 7XYU |
| Data collection |  |  |  |
| Space group | P $2_1$ | P $2_1$ | P $6_1$ |
| Cell dimensions |  |  |  |
| $a, b, c$ (Å) | 64.14, 137.19, 69.48 | 63.98, 138.27, 69.51 | 67.78, 67.78, 418.18 |
| α, β, γ (°) | 90.00, 117.51, 90.00 | 90.00, 117.38, 90.00 | 90.00, 90.00, 120.00 |
| Resolution (Å) | 45.83–1.70 (1.76–1.70)* | 138.27–1.50 (1.54–1.50) | 48.05–2.70 (2.80–2.70) |
| $R_{sym}$ or $R_{merge}$ | 0.096 (0.453) | 0.118 (1.568) | 0.016 (0.120) |
| $I / \sigma I$ | 13.33 (3.10) | 8.97 (1.59) | 33.30 (5.16) |
| Completeness (%) | 99.5 (98.7) | 99.9 (99.7) | 99.9 (99.9) |
| Redundancy | 6.9 (7.2) | 6.6 (5.4) | 2.0 (2.0) |
| Refinement |  |  |  |
| Resolution (Å) | 36.73–1.70 (1.76-1.70) | 34.61–1.50 (1.55–1.50) | 44.90–2.70 (2.80–2.70) |
| No. reflections | 116,008 (11450) | 170,645 (17008) | 29,648 (2992) |
| $R_{work} / R_{free}$ | 0.1466/0.1834 | 0.1693/0.2191 | 0.2503/0.3023 |
| No. atoms |  |  |  |
| Protein | 9586 | 7932 | 7827 |
| Water | 1640 | 1061 | 23 |
| $B$-factors |  |  |  |
| Protein | 15.6 | 25.1 | 84.6 |
| Water | 28.2 | 41.6 | 73.8 |
| R.m.s. deviations |  |  |  |
| Bond lengths (Å) | 0.007 | 0.006 | 0.009 |
| Bond angles (°) | 0.92 | 0.82 | 1.50 |

*Values in parentheses are for highest-resolution shell.

**Table 2 | Data collection and refinement statistics of ZYG11B**

|  | ZYG11B-SLFH | ZYG11B-AFLH | ZYG11B-CFLH |
|---|---|---|---|
| PDB accession code | 7XYV | 7XYW | 7XYX |
| Data collection |  |  |  |
| Space group | P $2_1 2_1 2_1$ | P $2_1 2_1 2_1$ | P $2_1 2_1 2_1$ |
| Cell dimensions |  |  |  |
| $a, b, c$ (Å) | 52.44, 97.07, 122.25 | 52.88, 97.27, 122.90 | 52.78, 97.37, 122.89 |
| α, β, γ (°) | 90.00, 90.00, 90.00 | 90.00, 90.00, 90.00 | 90.00, 90.00, 90.00 |
| Resolution (Å) | 76.02–2.52 (2.65–2.52)* | 61.45–2.50 (2.63-2.50) | 48.49–2.86 (2.96–2.86) |
| $R_{sym}$ or $R_{merge}$ | 0.076 (1.587) | 0.109 (1.413) | 0.259 (1.507) |
| $I / \sigma I$ | 18.40 (2.70) | 15.60 (2.70) | 26.00 (4.50) |
| Completeness (%) | 99.5 (99.8) | 92.2 (99.9) | 98.5 (95.0) |
| Redundancy | 13.1 (14.0) | 12.2 (13.0) | 13.3 (13.3) |
| Refinement |  |  |  |
| Resolution (Å) | 19.17–2.52 (2.61-2.52) | 20.49–2.50 (2.59-2.50) | 46.40–2.87 (2.97-2.87) |
| No. reflections | 21,715 (2150) | 20,888 (2204) | 14,868 (1399) |
| $R_{work} / R_{free}$ | 0.2172/0.2738 | 0.2304/0.2804 | 0.2020/0.2638 |
| No. atoms |  |  |  |
| Protein | 3950 | 3916 | 3899 |
| Water | 4 | 4 | 5 |
| $B$-factors |  |  |  |
| Protein | 101.4 | 80.5 | 74.2 |
| Water | 84.5 | 68.9 | 62.8 |
| R.m.s. deviations |  |  |  |
| Bond lengths (Å) | 0.004 | 0.010 | 0.010 |
| Bond angles (°) | 0.70 | 1.12 | 1.23 |

*Values in parentheses are for highest-resolution shell.

(Supplementary Fig. 3). The free α-amino group of the Nt-residue forms hydrogen bonds with the side chains of Asp556 and Asn597 in ZER1 (the counterparts of Asp 526 and Asn 567 in ZYG11B), respectively. Furthermore, the carbonyl oxygen of the Nt-residue is stabilized by two hydrogen bonds from Asn597 and Trp552 in ZER1 (Asp 567 and Trp 522 in ZYG11B) (Supplementary Fig. 3). The main-chain amide and carbonyl of the second residue (F2) on the peptide hydrogen bond to the main-chain respective carbonyl and amide of Asn679 in ZER1 (Ala647 in ZYG11B). Besides, a strong π-π interaction was formed between the aromatic side chain of F2 and the indole ring of Trp 552 in ZER1 (Trp 522 in ZYG11B) (Supplementary Fig. 3), this observation is consistent with our SPOT-array results that ZER1 and ZYG11B prefer an aromatic residue at position 2 (Fig. 1c, d). The main-chain amide of the third residue on the peptide is anchored by a hydrogen bond with Asn553 in ZER1 (Asn523 in ZYG11B) (Supplementary Fig. 3). It is worth noting that the side-chain hydroxyl group of Nt-Ser makes additional hydrogen bond with the side chain of Trp552 in ZER1 (Trp522 in ZYG11B) (Supplementary Fig. 3a, d), suggesting a critical role of the Trp552 in binding of Nt-Ser.

Apart from these direct interactions, a network of water molecules occupies the bottom of the cavity and mediates the contacts to ZER1 via water-mediated hydrogen bonds. In a close examination of the binding pockets of Nt-Ser (Ser1) and Nt-Gly (Gly1) in ZER1 (Fig. 3a), we found that there are two relatively conserved water molecules (water1 and water2 as shown in Fig. 3b, c) forming hydrogen-bonding bridges which connect the α-amino group of Ser1 or Gly1 with ZER1 Glu600. However, on the other side, the Water3 in the Ser1-binding mode has to move down and away from the corresponding Water3 in the Gly1-binding mode (-1Å) to provide enough space for docking the side chain of Ser1 (Fig. 3b, c). Simultaneously, the Asn633 shifts toward Water3 to form a hydrogen bond, thereby leaving no space for Water4 in the Ser1-binding mode. As a result, the Water3 mediates contacts directly with Water2, Gly593, Asn633 as well as Ser1, while in the Gly1-binding mode the Water3 together with Water4 establishes a contact network with Water2, Gly593, Asn633 as well as Trp552 (Fig. 3b, c). These rearrangements of water molecules might explain why ZER1 is able to recognize other Nt-residues in its narrow substrate-binding pocket. These structures may provide a useful guideline for the further design of more potent chemical probes against ZER1.

**Mutagenesis studies of ZER1 and ZYG11B**

To verify the functional importance of the residues involved in substrate binding, we generated single-point mutants of the binding pocket residues in ZER1 and ZYG11B. Our ITC assays indicated that any alanine mutation of W552, D556, N597 or E600 in ZER1, which would disrupt the direct hydrogen bonds with Nt-residue, abolished the binding to the XFLHVGQD (X = Ser, Ala, Cys or Thr) peptides (Fig. 3b–e and Supplementary Fig. 4a–c), highlighting the key role of these hydrogen bonds in mediating Nt-residue recognition. In addition, mutating the third-residue-interacting N553 into alanine caused a significant reduction in substrate binding (Fig. 3e, and Supplementary Fig. 4a–c). Likewise, consistent with the structural observations, the mutants W522A, D526A, N567A, E570A and N523A in ZYG11B were

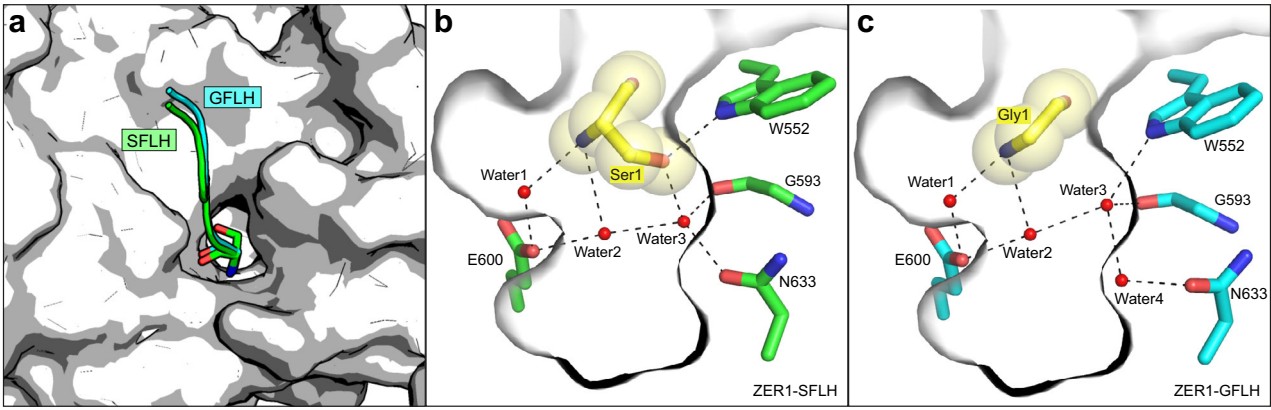

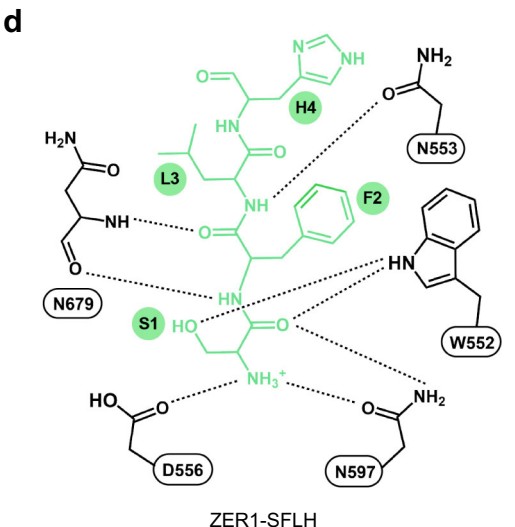

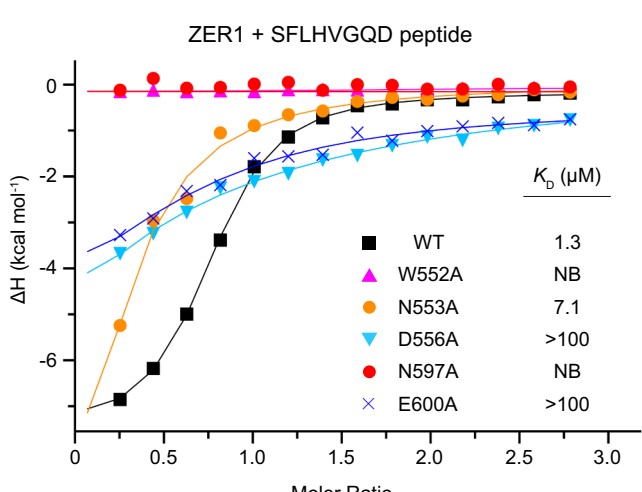

**Fig. 3 | Structural basis of Nt-Ser recognition by ZER1. a** Superposition of Nt-Ser and -Gly binding pocket in ZER1. Nt-Ser and -Gly bearing peptides (SFLH and GFLH) are shown in green and cyan, respectively. **b, c** Close-up view of the interactions of ZER1 with Nt-Ser (Ser1) and Nt-Gly (Gly1) in the binding pocket. The hydrogen bonds are shown as black dashed lines. **d** A schematic illustration of the recognition of SFLH peptide by ZER1. **e** ITC fitting curves of wild-type (WT) and mutant ZER1 (residues 469–766) titrated with SFLHVGQD peptide. The corresponding mutant proteins and binding affinities are indicated. NB, no detectable binding.

defective in binding to the XFLHVGQD (X = Ser, Ala or Cys) peptides (Supplementary Fig. 4d–f). These results suggest that ZER1 and ZYG11B utilize a conserved binding pocket composing a set of hydrogen bonds to capture either the Nt-Gly or small non-Gly Nt-residues.

To further substantiate the role of these interacting residues in cells, we mutated the W552 or N597 to alanine in the Flag-tagged full-length ZER1. Then we performed Co-IP experiments in NAA10-KD cells to test the interactions of the ZER1 mutants with target proteins. The results showed that these mutations greatly impaired the interactions with Ser-GFP or Ala-GFP fusion protein (Fig. 4a, b). Correspondingly, GPS assays showed that unlike WT ZER1, overexpression of the W552A or N597A mutant did not efficiently target the Nt-Ser, -Ala or -Cys-GFP fusion proteins for degradation (Fig. 4c–j).

To test whether ZER1/ZYG11B was able to target native proteins starting with Nt-Ser in the human proteome, we analyzed the interactors of ZER1 and ZYG11B using BioGRID database[23] and found the ACHAP (acetylcholinesterase-associated protein) bearing an Nt-Ser may be a potential substrate. In fact, overexpression of ZER1 or ZYG11B caused a reduction in Flag-tagged ACHAP levels (Supplementary Fig. 5a, b). Furthermore, in contrast to WT ZER1, the mutant W552A or N597A failed to induce the ACHAP degradation (Supplementary Fig. 5c, d). Thus, ZER1 and ZYG11B might have a broad substrate spectrum in cells.

In conclusion, we propose that ZER1/ZYG11B not only targets Gly/N-degrons for proteasomal degradation, but it also has the ability to bind a defined selection of non-Ac/N-degrons by a conserved molecular mechanism of substrate recognition and target their degradation through the non-Ac/N-degron pathway.

## Discussion

Intracellular protein levels are elaborately regulated by the ubiquitin-proteasome system (UPS). The N-degron pathway is a proteasome-dependent proteolytic system that eliminates the aberrant and unnecessary proteins based on their destabilizing Nt-residues[24,25]. Several branches of the N-degron pathways have been established in eukaryotes, such as the Arg/N-degron pathway[24], the Ac/N-degron pathway[8], the Pro/N-degron pathway[26] and the Gly/N-degron pathway[19]. In these N-degron pathways, dedicated E3 ligases (N-recognins) are responsible for selective degradation by targeting specific Nt-residues of substrates.

Of particular interest is that different N-recognins are likely to target several kinds of Nt-residues other than an exclusive Nt-residue, although they may prefer a certain residue. For instance, in the Arg/N-degron pathway, UBR1 prefers Nt-Arg but also allows -Lys and -His docking in the UBR box[27]. In addition, the ClpS-homology domain of UBR1 is able to recognize several bulky hydrophobic residues

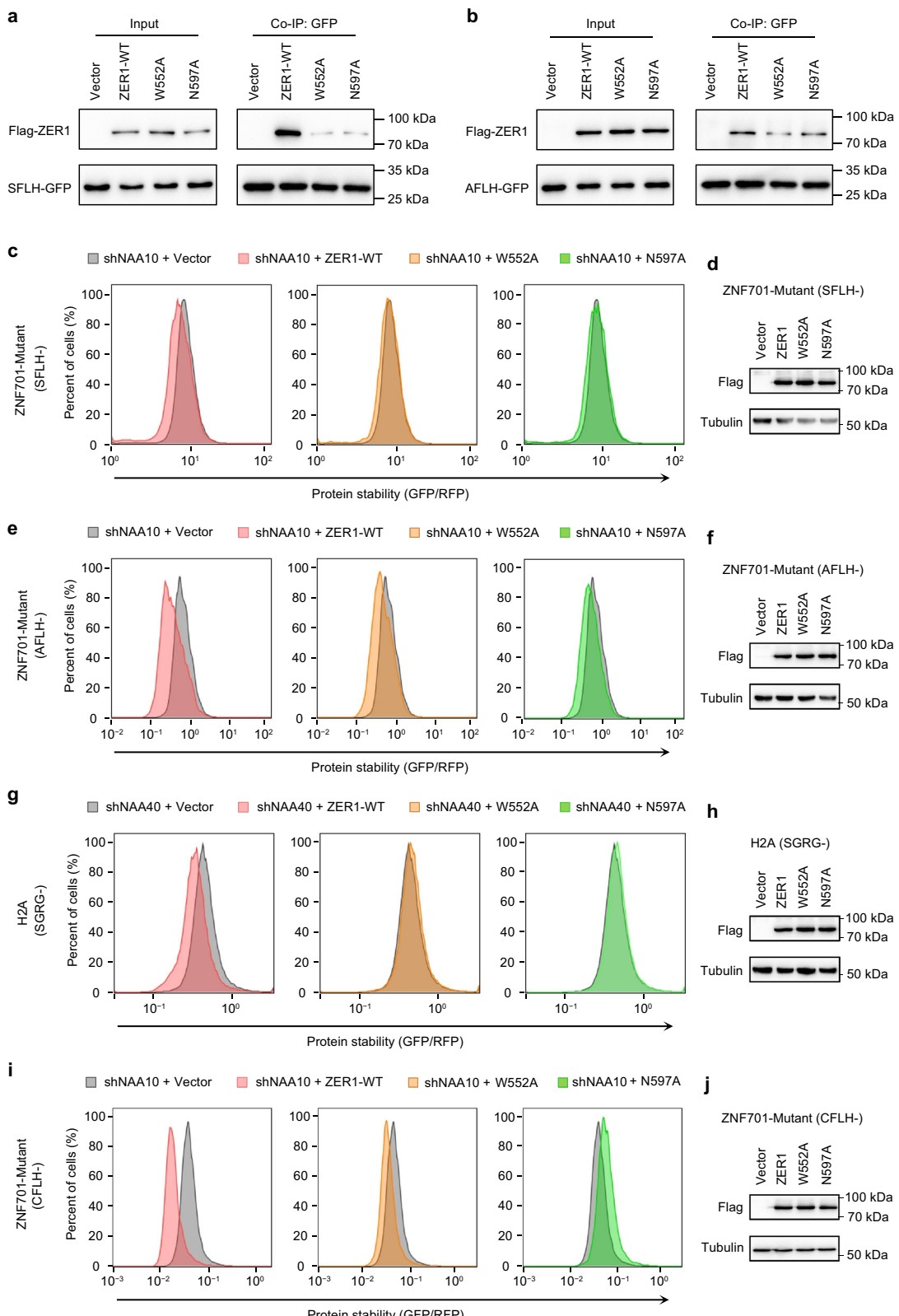

(Nt-Trp, -Phe, -Tyr, -Leu and -Ile) as well as proteins bearing an Nt-Met followed by a bulky hydrophobic residue[28–30]. In the Pro/N-degron pathway, human GID4 was thought to exclusively bind Nt-Pro, but subsequent studies revealed that GID4 can also bind Nt-hydrophobic residues including -Ile, -Leu, -Val or -Phe[26,31–34]. Herein, we found in biochemical and cellular experiments that are supported by crystal structures that ZER1 and ZYG11B, the N-recognin of the Gly/N-degron

pathway, can recognize not only Nt-Gly but other small Nt-residue as well largely expanding the range of N-degrons. Specifically, ZER1 binds to Nt-Ser, -Ala, -Thr or -Cys with higher affinity than the Nt-Gly in the otherwise identical sequence context. The Nt-Ser, -Ala, or -Cys can be bound by ZYG11B as well.

Of note, small Nt-residues (such as -Ser, -Ala, -Thr, -Val or -Cys) are the canonical substrates of NatA, and the resulting Nt-Ac state has been

**Fig. 4 | Mutational analysis of degron-binding residues of ZER1. a** Co-IP analysis of the interactions of WT full-length ZER1 and indicated mutants with SFLH-fused GFP in NAA10 KD cells. HEK293T cells expressing SFLH-fused GFP were infected with shNAA10 lentivirus and selected for stable cells. The stable cells were transfected with Flag-tagged wild-type and mutant ZER1. Cells were treated with MG132 (10 μM) for 6 h at 48 h post-transfection and then harvested for Co-IP using anti-GFP beads and analyzed by western blotting. Representative images, $n$ = 3. **b** Co-IP analysis of the interactions of WT full-length ZER1 and indicated mutants with AFLH-fused GFP. Representative images, $n$ = 3. **c** Stability analysis of SFLH-fused GFP with overexpression of WT and mutant ZER1 proteins in NAA10 KD cells. The ratio of GFP/RFP was analyzed by flow cytometry. **d** Western blot analysis of WT and mutant ZER1 expression in SFLH-GFP reporter cell lines. Representative images,

$n$ = 3. **e** Stability analysis of AFLH-fused GFP with overexpression of WT and mutant ZER1 proteins in NAA10 KD HEK293T cells. **f** Western blot analysis of WT and mutant ZER1 expression in AFLH-GFP reporter cell lines. Representative images, $n$ = 3. **g** Stability analysis of SGRG-fused GFP with overexpression of WT and mutant ZER1 proteins in NAA40 KD HEK293T cells. **h** Western blot analysis of WT and mutant ZER1 expression SGRG-GFP reporter cell lines. Representative images, $n$ = 3. **i** Stability analysis of CFLH-fused GFP with overexpression of WT and mutant ZER1 proteins in HEK293T cells. **j** Western blot analysis of WT and mutant ZER1 expression in CFLH-GFP reporter cell lines. Representative images, $n$ = 3. Uncropped western blot images are provided in Source data 1. All FACS sequential gating images are provided in Supplementary Fig. 6.

previously established as the Ac/N-degron conditionally targeted for degradation through the Ac/N-degron pathway[8,9]. However, the concept of non-Ac/N-degrons created by these small residues is currently emerging. One study showed that the non-Ac peptides derived from NatA substrates efficiently bind to E3 ubiquitin ligases of the IAP (inhibitor of apoptosis protein) family in cell lysates and that they are ubiquitinated by IAPs in vitro, suggesting that Nt-Ac generally blocks protein degradation in human cells[15]. In parallel, depletion of NatA activity was shown to trigger protein destabilization of NatA substrates, suggesting a non-Ac/N-degron marks proteins for degradation via the UPS in plants[16]. Although the clear homologs of IAPs or CRL2$^{ZER1/ZYG11B}$ are missing in Arabidopsis, the neddylation of Cullin 1 is significantly enhanced in NatA depleted plants[16]. We report here that ZER1 is able to recognize Nt-Ser or -Ala for protein degradation in NatA defective cells, it seems therefore that the Nt-Ac modification is a conserved protective function in the protein stabilization[35]. It is worth to note that the BIR domains of IAPs prefer Nt-Ala over other residues[36] and the replacement of Nt-Ala with -Ser caused a loss of activity[37]. In contrast, we show here that ZER1 and ZYG11B strongly bind to Nt-Ser, -Ala, -Cys as well as -Gly suggesting that IAPs and ZER1/ZYG11B may selectively target context-specific degron for degradation, although they all recognize mainly the first four residues of their substrates. This present study will facilitate further discovery of more physiological substrates of ZER1 and ZYG11B.

On the other hand, all 20 natural amino acids have been characterized as destabilizing Nt-residues, giving the functional crosstalk between the N-degron pathways[38]. For example, proteins bearing MΦ (Met followed by a bulky hydrophobic residue such as Leu, Ile, Phe, Tyr, Trp) can be targeted for degradation by the Arg/N-degron pathway. Alternatively, if the Nt-Met in this context is acetylated by NAT, the resultant degron is subject to the Ac/N-degron pathway, Rgs2 is one such physiologically relevant example[39–42]. Likewise, small Nt-residues can become N-degrons either in the Nt-Ac or non-Ac version. The Ac/N-degron is recognized by TEB4 (the human ortholog of yeast Doa10)[8], the latter can be selectively recognized by ZER1/ZYG11B as shown here. In the physiologically relevant context, the irreversible cotranslational Nt-Ac modification may confer protein stabilization by proper folding and assembly into a cognate complex. Otherwise, the exposed Ac/N-degrons will be eliminated by the Ac/N-degron pathway. Another scenario is that if proteins failed to undergo Nt-acetylation by NatA, the free Nt-residues can be recognized by IAPs or CRL2$^{ZER1/ZYG11B}$ for degradation, mirroring the CRL2$^{ZER1/ZYG11B}$ –mediated "myristoylation quality control". Therefore, CRL2$^{ZER1/ZYG11B}$ and IAPs, along with NatA may cooperatively safeguard cellular protein homeostasis in a spatial and temporal manner.

## Methods
### Peptide array binding assays
The regions encoding the ZER1 (residues 469–766) and ZYG11B (residues 490–744) were cloned into the pACYC-GST (Addgene Plasmid #62329) vector. Binding experiments of ZER1$_{469-766}$ and ZYG11B$_{490-744}$ to peptide arrays were conducted as described[34]. In brief, peptide

arrays were synthesized on cellulose membrane using an Autospot Multipep system (Intavis). The peptide arrays were blocked overnight at 5% milk-PBST solution at 4 °C, 1 × PBST is 0.05% Tween 20, 0.14 M NaCl, 2.7 mM KCl, 4.3 mM Na$_2$HPO$_4$, 1.4 mM KH$_2$PO$_4$, pH 7.5. After three washings with PBST, the peptide array was preincubated for 5 min in interaction buffer (10% glycerol, 0.1 M KCl, 1 mM EDTA, 20 mM HEPES pH 7.5). The array was then incubated for 1 h at room temperature in the interaction buffer containing purified GST-tagged ZER1$_{469-766}$ (50 nM) or ZYG11B$_{490-744}$ (230 nM). The membrane was washed three times with PBST, followed by incubation 1 h at room temperature with the primary antibody to GST (1: 6,000, anti-GST, polyclonal; GE Healthcare, cat. no. 10595345). After washing, the membrane was incubated for 1 h at room temperature with secondary antibody (1: 12,000, anti-goat IgG [whole-molecule]-peroxidase antibody produced in rabbit; Sigma-Aldrich, cat. No. A4174). The signal was detected by chemiluminescence using a Hyperfilm TM high-performance autoradiography film (GE Healthcare). All assays were carried out twice. The data were normalized and the average binding of ZER1 or ZYG11B to each peptide was calculated, and a sequence logo was prepared using WebLogo 3 (http://weblogo.threeplusone.com/).

### Protein expression and purification
Plasmid constructions were performed as previously described with minor modifications[20]. Briefly, the coding sequence of ZYG11B (residues 485–728) was cloned into a modified pET28-MKH8SUMO (Addgene Plasmid #79526) vector engineered with an N-terminal 8 × His-SUMO fusion tag followed by a TEV (Tobacco Etch Virus) cleavage site (LYFQY ↓ X, X = A, S or C) and FLH polypeptide. The N-terminal residue (A, S or C) can be exposed by the TEV protease during protein purification. The SFLH or AFLH peptide fused ZER1 (residues 518–766) and TFLH peptide fused ZER1 (residues 521–766) were obtained by a similar approach.

The fusion protein was expressed in *E. coli* strain BL21 (DE3) overnight at 18°C after induction with 0.2 mM isopropyl β-D-1-thiogalactopyranoside (IPTG). All the fusion proteins were purified by standard Ni-NTA affinity chromatography, treated with TEV protease in dialysis buffer (20 mM Tris-HCl pH 7.5, 300 mM NaCl and 2 mM β-mercaptoethanol) at 4 °C overnight. The products were passed through a Ni-NTA column to remove the 8 × His-SUMO tag and TEV protease. The target protein was further purified using a HiTrap Q HP column (GE healthcare) and a Superdex 200 Increase 10/300 GL (GE healthcare) with buffer containing 20 mM Tris-HCl pH 7.5, 150 mM NaCl and 1 mM DTT. Finally, the target protein was then concentrated and stored at −80 °C for later use.

### Protein crystallization
Proteins were crystallized using the sitting-drop vapor diffusion method. Briefly, 1 μL protein and 1 μL reservoir solution were mixed and set up in 48-well crystallization plates at 18°C. The crystallization conditions were as follows: SFLH-fused ZYG11B$_{485-728}$, AFLH-fused ZYG11B$_{485-728}$ and CFLH-fused ZYG11B$_{485-728}$ were crystallized in 0.1 M MES (pH 6.5) and 1.6 M MgSO$_4$; SFLH-fused ZER1$_{518-766}$ and AFLH-fused

ZER1$_{518-766}$ were crystallized in 0.2 M NH$_4$F and 20% (w/v) PEG3350; TFLH-fused ZER1$_{520-766}$ was crystallized in 0.16 M Sodium Malonate (pH 7.0) and 20% (w/v) PEG 3350. All crystals were soaked in a cryo-protectant containing the reservoir solution plus 25% (v/v) glycerol before flash freezing in liquid nitrogen for data collection.

### Data collection and structure determination
X-ray diffraction data were collected on the beamline BL02U or BL19U at Shanghai Synchrotron Radiation Facility (SSRF) or beamline BL-1A at the Photon Factory, KEK, Japan. The data were processed and scaled using XDS[43]. The structures of ZYG11B or ZER1 were solved by molecular replacement with Phaser[44] using PDB entry 7EP1 and 7EP3 structures as search templates. The initial models were then optimized through several rounds of manual model rebuilding in Coot[45] and refinement with PHENIX[46]. Structural figures were generated by PyMOL program (https://www.pymol.org).

### Isothermal titration calorimetry
ZYG11B (residues 490–728) bearing an N-terminal SELF generated by a modified pET28-MKH8SUMO (Addgene Plasmid #79526) vector and ZER1 (residues 469–766) bearing an N-terminal MEEL sequence generated by pNIC-CH (Addgene Plasmid #26117) vector were used to perform isothermal titration calorimetry (ITC) assays, as previously described[20]. Proteins (concentration 50–70 μM) and peptides (concentration 0.5–1.5 mM) solutions were prepared in an ITC buffer consisting of 20 mM Tris-HCl (pH 7.5) and 150 mM NaCl. The ITC measurements were recorded at 16 °C with a MicroCal PEAQ-ITC instrument (Malvern Panalytical). 1.5 μL of each peptide (except for first injection using 0.5 μL peptide) was titrated into a 300 μL cell of the protein with 120 s incubation time between the injections. The reference power was controlled at 10 mcal sec$^{-1}$. The acquired data were analyzed using MicroCal PEAQ-ITC Analysis Software version 1.30. Each experiment was repeated at least twice independently with similar results.

### Global protein stability (GPS) assay
The oligonucleotide encoding the 12-mer peptide from N-terminus of ZNF701 or H2A was cloned into pCDH-Ub-MCS-GFP-IRES-RFP vector using the Seamless Cloning method (Beyotime, D7010S). The GPS reporter vector (pCDH-Ub-MCS-GFP-IRES-RFP) was constructed based on pCDH-Flag-c-Myc (Addgene Plasmid #102626) by adding Ub-MCS-GFP-IRES-RFP cassette and changing puromycin-resistance gene to blasticidin S-resistance gene. First, the GPS reporter stable cells were constructed by lentiviral transduction. Then wild-type and mutant ZER1 were overexpressed in GPS reporter cells or NAA10 was knocked down using pLKO.1 shRNA lentivirus. Last, the stability of peptide-GFP was determined by measuring the cellular GFP/RFP ratio through flow cytometry using RFP as an internal control. Cells were analyzed by flow cytometry using an ACEA NovoCyte instrument (ACEA Biosciences, Inc) and the data was analyzed using FlowJo. Expression of wild-type and mutant ZER1 in GPS reporter cells were examined by western blot with anti-Flag (1:1,000; Proteintech, 20543-1-AP) and anti-alpha tubulin (1: 5,000; Beyotime, AF0001) was used as an internal control. All uncropped blots are provided in Source data 1.

### RNA interferences and real-time PCR
For individual gene knockdown experiments, the pLKO.1-puro vector was used. Oligonucleotides encoding the sense and antisence strands of shRNAs were synthesized, annealed and cloned into the pLKO.1-puro vector. The shRNA sequences were: shNAA10: GCCATGATAGAGAACTTCAAT; shNAA40: GAAGCGTTGCAATTTGAAATT. Total RNA was extracted using an RNeasy plus kit (Qiagen) and reverse-transcribed using an iScrip reverse transcription kit (Bio-Rad). Quantitative real-time PCR (qPCR) analyses were performed using Power SYBR Green PCR Master Mix and the ABI 7500-FAST Sequence Detection System (Applied Biosystems). Gene expressions were

calculated following normalization to GAPDH levels using the comparative Ct (cycle threshold) method. Statistic differences were calculated using a two-way unpaired Student's $t$ test. The paired primers sequences for qPCR were: qPCR-NAA10: 5'-TGCTGAGGACGAGAATGGGAAG-3' and 5'-CTGGTCCATCAGTTTCTGAGCC-3'; qPCR-NAA40: 5'-CGTTTGTGCCAAAGTGGACGCT-3' and 5'-TCCAGTCCAGACACTCGCTTAC-3'. Source data for qPCR results are provided in Source data 2.

### Cell culture, transfection, lentiviral production and infection
Human HEK293T (ATCC CRL-3216) cell line was maintained in DMEM medium (Cellgro) supplemented with 10% FBS (BioInd) and 1% penicillin/streptomycin. HEK293T cells were co-transfected with pMD2.G (Addgene Plasmid #12259), psPAX2 (Addgene Plasmid #12260), and pLKO.1 shRNA, lentiCRISPR v2 gRNA or pCDH tagged-cDNA constructs. After 2 days, lentiviral supernatants were collected for infection. Cells were incubated with lentiviral supernatants in the presence of 8 μg/mL polybrene. After 48 h, the infected cells were selected with puromycin (2 μg/mL) or blasticidin (10 μg/mL) for indicated pLKO.1/lentiCRISPR v2/pCDH stable clones for at least 3 days before experiments. Human ZER1 cDNA (NCBI Gene ID 10444) was cloned into pCDH-Flag-c-Myc (Addgene, #102626) using Seamless Cloning method. The sgRNAs sequences used for CRISPR/Cas9-mediated gene silencing were: sgNT: CACCGGGATACTTCTTCGAACGTTT; sgZYG11B: GCGCTCGTAAGGATCCTCGA; sgZER1: GCCGCAGCAGGGACTCCACA.

### Co-immunoprecipitation (Co-IP)
HEK293T cells stably expressing the indicated epitope-tagged proteins were grown in 10 cm cell culture plates. Treatment with or without MG132 (10 μmol/mL) before cells collection was indicated in figure legends. For Co-IP, cells were lysed in ice-cold cell lysis buffer (50 mM Tris-HCl pH 7.4, 250 mM NaCl, 0.5% Triton X100, 10% glycerol, add freshly 1 mM DTT, 1 mM PMSF, and 1× protease inhibitors). GFP-Trap® Magnetic Particles (Proteintech) were added to the supernatants and incubated with rotation overnight at 4 °C. The beads were then washed four times with the same lysis buffer before bound proteins were eluted using SDS-PAGE sample buffer (95 °C, 5 min). Proteins were subsequently resolved by SDS-PAGE and transferred to a PVDF membrane (EMD Millipore). The chemiluminescent imaging was produced by 5200CE Tanon™ Chemi-Image System. The antibodies of anti-Flag (1: 1,000; Proteintech, 20543-1-AP), anti-GFP (1: 5000 Proteintech, 50430-2-AP) and anti-alpha tubulin (1: 5,000; Beyotime, AF0001) were used for immunoblot.

### Mass spectrometry
HEK293T cells expressing Ub-SFLH-GFP were infected with shNT and shNAA10 lentivirus and selected for stable cells. The stable cells were collected and lysed in ice-cold cell lysis buffer (50 mM Tris-HCl pH 7.4, 250 mM NaCl, 0.5% Triton X100, 10% glycerol, add freshly 1 mM DTT, 1 mM PMSF, and 1x protease inhibitors). GFP-Trap® Magnetic Particles (Proteintech) were added to the supernatants and incubated with rotation overnight at 4 °C. The beads were then washed four times with the same lysis buffer before bound proteins were eluted using SDS-PAGE sample buffer (95 °C, 5 min). Proteins were subsequently resolved by SDS-PAGE. Protein bands were excised from the gel and cut into small pieces (1 mm).

After desalting, the digested samples were injected into a Nano-LC system (EASY-nLC 1000, Thermo Fisher Scientific). Samples were run on a C18 column (50 μm inner-diameter×15 cm, 2 μm C18) at a flow rate of 300 nL/min. The HPLC gradient was: 5% to 13% solvent B (0.1% formic acid plus acetonitrile) in 6 min, 13% to 28% solvent B in 15 min, 28% to 45% solvent B in 8 min and 45% to 100% Solvent B for 1 min, 100% Solvent B for 5 min. The HPLC eluate was sprayed directly onto an Orbitrap Q-Exactive mass spectrometer (Thermo Fisher Scientific). The power supply operating voltage is 2.4 kV. Mass spectrometry analysis was performed in a data-dependent mode. MS1

measurement scans were performed using an automatic gain control (AGC) objective of 1e6 and a resolution of 70,000. MS2 spectral resolution was 17,500.

Data analysis was performed using Proteome Discoverer software 1.4. The minimum protein identification probability was set at two unique peptides with ≤1.0% FDR. The minimum peptide length was 6. Proteins with scores below 2 were removed. Peptide sequences were screened exclusively using trypsin and up to two missed digests were tolerated. Aminomethylation on cysteine has been designated as a fixed modification. Methionine oxidation, N-terminal acetylation and lysine were used as variable modifications. Precursor mass tolerances were adjusted to ±10 ppm precursor and ±0.02 Da MS/MS.

### Reporting summary

Further information on research design is available in the Nature Portfolio Reporting Summary linked to this article.

### Data availability

Atomic coordinates for the reported structures have been deposited in the Protein Data Bank (PDB) under accession codes 7XYS for ZER1-SFLH, 7XYT for ZER1-AFLH, 7XYU for ZER1-TFLH, 7XYV for ZYG11B-SLFH, 7XYW for ZYG11B-AFLH, and 7XYX for ZYG11B-CFLH. The mass spectrometry proteomics data have been deposited to the ProteomeXchange Consortium via the PRIDE partner repository with the dataset identifier PXD037312. All data needed to evaluate the conclusions in the paper are present in the paper and/or the Supplementary Information.

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

## Acknowledgements

We thank the staff at beamlines BL02U and BL17B of the Shanghai Synchrotron Radiation Facility and the staff at beamline BL-1A at the Photon Factory, KEK, Japan for assistance with X-ray data collection. We thank Kai Zhang and Xue Bai for assistance with mass spectrometry. This work was supported by Shandong Province Special Fund "Frontier Technology and Free Exploration" from Pilot National Laboratory for Marine Science and Technology (Qingdao) (No. 8-01), National Natural Science Foundation of China grants 32271265 (to C.D.), 31900865 (to C.D.), 32071193 (to C.D.), 81974431 (to W.M.), 82173000 (to W.M.), 81874039 (to X.Y.) and 82103176 (to X.W.), National Youth Top-Notch Talent Support Program in China, and Research Foundation of Tianjin Municipal Education Commission, China (2021ZD036).

## Author contributions

C.D. and W.M. conceptualized the project and designed experiments. S.W. and A.J. conceptualized and performed the peptide array binding assays. Y.L., Y.Z. and X.Y. cloned the constructs and carried out protein purification, crystallization, GST pull-down, and ITC assays. X.Y. determined the crystal structures and analyzed the data. Y.Z., C.Y. and X.W. performed the Co-IP, GPS experiments and LC-MS assays with the help from L.S. and C.J. C.D., B.Z. wrote the manuscript with critical inputs from all authors.

## Competing interests

The authors declare no competing interests.
