## [Peer Review File · Nature Communications]

REVIEWER COMMENTS

Reviewer #1 (Remarks to the Author):

In the paper entitled "CRL2ZER1 recognizes small N-terminal residues for degradation", Dong and colleagues describes an expanded substrate specificity for ZER1 and ZYG11B, the substrate receptors of CRL2 with implication for quality control of proteins that fail to undergo proper Nt-acetylation. While most endogenous substrates of ZER1 and ZYG11B are proteins starting with glycine, the authors found that ZER1/ ZYG11B are able to interact with other small Nt amino acids such as alanine, serine and threonine, sometimes with an even stronger affinity than glycine. Structural approaches defined the molecular mechanism of recognition of these small Nt residues. Complemented with in vitro binding assay and GPS assays the critical residues important for recognition were mapped. Interestingly, in vivo overexpression of ZER1/ ZYG11B did not induce degradation of Ala-GFP or Ser-GFP starting peptides unless Nt-acetylation is inhibited. This suggests a protective role for Ala/ser Nt-acetylation, preventing recognition by ZER1/ ZYG11B. Given the large proportion of proteins starting with small amino acids such as Ala/ser, Nt-acetylation might play an important evolutionary role of protecting these proteins from rapid degradation by ZER1/ ZYG11B. Alternatively, ZER1/ ZYG11B might function in a Nt-acetylation quality control, degrading proteins that fail to undergo Nt-acetylation.

Overall the authors did excellent job in incorporation of various experimental approaches (structural, biochemical, cell based) to map the molecular mechanism of recognition. Very elegant co-immunoprecipitation experiments using various ZER1/ZYG11B mutants supported their findings.

Manuscript is well written and very clear. Data presented is reliable and of high quality. The importance of a non-modified Nt residue to ZER1/ ZYG11B binding was clearly proven.

A few points of weakness I found in the manuscript:

a. I found the GPS assays done in N-terminal acetylation silenced cells less convincing, with the magnitude of stability changes of the various substrates very minor even upon overexpressing ZER1/ ZYG11B.

b. Substrate degradation was monitored in cells overexpressing ZER1/ZYG11B and silenced for N-terminal acetylation both are non physiological conditions. As N-terminal acetylation is very efficient and non-reversible, it is not clear to me what are the biological conditions in which ZER1/ZYG11B operate to degrade proteins that fail to undergo N-terminal acetylation.

Unlike Nt-Glycine that is hardly acetylated and thus serves as a potent and "real" N-degron in protein substrates, Nt-Ala/Ser/Thr (although potentially could be recognized by ZER1/ ZYG11B) might not be physiological relevant and "real" substrates.

c. Assays carried using short peptides nicely support that ZER1/ZYG11B can interact with small amino acids other than glycine. However, evidence is still missing to show that full length proteins containing Nt Ala/Ser are also substrates of ZER1/ZYG11B.

Specific experiments to strengthen the manuscript and address these concerns:

1. Ub-GPS experiments with NAA10 silencing needs to be done in combination with ZER1/ZYG11B knockout or knockdown. We expect stabilization of Ala/Ser-GFP peptides in NAA10+ ZER1/ ZYG11B ablated cells compared to NAA10 KD alone.

2. Evidence for full length substrates of ZER1/ZYG11B starting with Ala/Ser need to be provided. This can be done by an educational guess of known Nt Ala/Ser proteins that have a perfect ZER1/ZYG11B degron consensus and might be potential substrates. Alternatively, mass spec experiment of NAA10 silenced cells should recover endogenous substrates whose protein levels are decreased as a result of N-terminal acetylation inhibition.

Minor comment:

Experiments might work better in siRNA (and not shRNA) NAA10 depleted cells. The knockdown achieved with shRNA is very good (Fig. S2D), however the acetylation remained high and hardly changed (Fig. S2C), which might suggest compensatory mechanism activated in NAA10 KD cells.

Reviewer #2 (Remarks to the Author):

In the manuscript "CRL2/ZER1 recognizes small N-terminal residues for degradation" by Li et al., the authors have examined the specificity of the Cullin2 RING E3 ligase (CLR2) substrate receptors (namely ZER1 and ZYG11B).

The authors used a range of biochemical and structural approaches to demonstrate that both ZER1 and ZYG11B have an extended specificity, beyond the original defined specificity (i.e. recognition of N-terminal (Nt) Gly). Here they have focused largely on Nt-Ser and Nt-Ala, although they also clearly demonstrate that Nt-Thr and Nt-Cys are also recognised by ZER1, similarly Nt-Cys is also recognised by ZYG11B. Although the recognition of Nt-Cys (by ZER1 and ZYG11B) was not initially defined using the peptide array (because of the limitation of this method), the identification of Nt-Cys as a primary destabilising residue (primary degron) in vitro, is very interesting. To date, Nt-Cys has only been identified as a secondary destabilising residue (following oxidation) and attachment of a primary destabilising residue. Therefore, I believe that the identification of Nt-Cys as a primary destabilising residue should be further examined/validated in cells, using the GPS assays, in normal and NatA deficient cells.

Minor comments

Regarding the title, the authors only mention ZER1, even though the manuscript examines both CLR2 receptors (ZER1 and ZYG11B) equally. The authors should consider a change to the title to reflect the full extent of their data.

There is a misuse of articles (i.e. the) throughout the manuscript e.g. "degradation in the NatA-deficient cells", should not include "the". "Quality control through Gly/N-degron pathway", should include "the" before Gly/N-degron. Likewise, "Through Ac/N-degron pathway", should include "the" before Ac/N-degron.

There are several references throughout the manuscript describing the binding of ZER1/ZYG11B to non-Ac/N-degrons, the authors need to be more precise with the terminology as ZER1/ZYG11B does not bind to all non-Ac/N-degrons, only a selection of non-Ac/N-degrons.

Line 48, as the modification occurs to the N-terminus, and is not facilitated by the N-terminus the following sentence on line 48 "The resulting Nt-Ac changes the chemical properties of the N-terminus by neutralizing its positive charge, and generally prevents the N-terminus from other modifications." should be revised to something like "prevents other modifications of the N-terminus"

At several places in the manuscript the authors appear to avoid discussing the recognition of Nt-Cys (and Nt-Thr). E.g. Line 99 – the authors identified that ZER1 binds Nt-Ser, Nt-Thr and Nt-Ala in preference to Nt-Gly, however they do not mention Nt-Thr here – even though is it clearly shown (and discussed) later in the manuscript. Again on Line 116 – the authors focus only on Nt-Ser and Nt-Ala, although both Nt-Thr and Nt-Cys (in the case of ZER1) exhibit stronger binding than Nt-Gly.

The authors should adjust the text to include all their data.

Line 209 D556A should read D556

The discussion contains a few imprecise/incomplete statements/definitions that should be revised.

The Arg/N-degron pathway (via UBR1) is not only responsible for the recognition of type 1 residues Nt-Arg, -Lys and -His (via the UBR box) but also responsible for the recognition of type 2 residues (Nt-Trp, -Phe, -Tyr, -Leu and -Ile) as well as proteins bearing an Nt-Met-hy sequence (via the ClpS domain site within UBR1) (Xia et al., 2008; Schuenemann et al., 2009; Kim et al., 2013).

In addition to the recognition of Nt-Pro, -Ile, -Leu and -Phe, GID4 is able to recognise Nt-Val.

Line 237 the following statement “the N-recognin is likely to target variable Nt-residues” should be revised, as the N-recognin does not target variable residues, rather it targets a defined selection of specific Nt-residues

Response to Reviewer #1

Comments:

In the paper entitled “CRL2^{ZER1} recognizes small N-terminal residues for degradation”, Dong and colleagues describes an expanded substrate specificity for ZER1 and ZYG11B, the substrate receptors of CRL2 with implication for quality control of proteins that fail to undergo proper Nt-acetylation. While most endogenous substrates of ZER1 and ZYG11B are proteins starting with glycine, the authors found that ZER1/ ZYG11B are able to interact with other small Nt amino acids such as alanine, serine and threonine, sometimes with an even stronger affinity than glycine. Structural approaches defined the molecular mechanism of recognition of these small Nt residues. Complemented with in vitro binding assay and GPS assays the critical residues important for recognition were mapped. Interestingly, in vivo overexpression of ZER1/ ZYG11B did not induce degradation of Ala-GFP or Ser-GFP starting peptides unless Nt-acetylation is inhibited. This suggests a protective role for Ala/ser Nt-acetylation, preventing recognition by ZER1/ ZYG11B. Given the large proportion of proteins starting with small amino acids such as Ala/ser, Nt-acetylation might play an important evolutionary role of protecting these proteins from rapid degradation by ZER1/ ZYG11B. Alternatively, ZER1/ ZYG11B might function in a Nt-acetylation quality control, degrading proteins that fail to undergo Nt-acetylation.

Overall the authors did excellent job in incorporation of various experimental approaches (structural, biochemical, cell based) to map the molecular mechanism of recognition. Very elegant co-immunoprecipitation experiments using various ZER1/ZYG11B mutants supported their findings.

Manuscript is well written and very clear. Data presented is reliable and of high quality.

The importance of a non-modified Nt residue to ZER1/ ZYG11B binding was clearly proven.

A few points of weakness I found in the manuscript:

a. I found the GPS assays done in N-terminal acetylation silenced cells less convincing, with the magnitude of stability changes of the various substrates very minor even upon overexpressing ZER1/ ZYG11B.

b. Substrate degradation was monitored in cells overexpressing ZER1/ZYG11B and silenced for N-terminal acetylation both are non physiological conditions. As N-terminal acetylation is very efficient and non-reversible, it is not clear to me what are the biological conditions in which ZER1/ZYG11B operate to degrade proteins that fail to undergo N-terminal acetylation.

Unlike Nt-Glycine that is hardly acetylated and thus serves as a potent and “real” N-degron in protein substrates, Nt-Ala/Ser/Thr (although potentially could be recognized by ZER1/ ZYG11B) might not be physiological relevant and “real” substrates.

c. Assays carried using short peptides nicely support that ZER1/ZYG11B can interact

with small amino acids other than glycine. However, evidence is still missing to show that full length proteins containing Nt Ala/Ser are also substrates of ZER1/ZYG11B.

Response:

Thank you very much for your positive comments and constructive suggestions to our manuscript. We supplied the requested experiments you mentioned to strengthen the manuscript and address these concerns as follows.

Specific experiments to strengthen the manuscript and address these concerns:

Major comments:

1. Ub-GPS experiments with NAA10 silencing needs to be done in combination with ZER1/ZYG11B knockout or knockdown. We expect stabilization of Ala/Ser-GFP peptides in NAA10+ ZER1/ ZYG11B ablated cells compared to NAA10 KD alone.

Response:

We thank the reviewer for the great suggestion. We followed the reviewer's suggestion and performed Nt-Ser degron GPS experiments with NAA10 silencing in combination with ZER1/ZYG11B knockout. Ser-GFP was destabilized upon NAA10 knockdown, and partial restabilization was observed upon ablation of both NAA10 and ZER1/ZYG11B compared to NAA10 alone (Fig. R1a). It is possible that there are additional E3 ligase(s) besides ZER1/ZYG11B that can recognize the non-Ac Nt-Ser degron as well.

Considering that the Nt-Ac levels of Ser-GFP remained constantly high even in NAA10-KD cells, we used Cys-GFP which is known to be only 50% Nt-acetylated by NatA to perform the GPS again. Indeed, Cys-GFP was destabilized upon NAA10 knockdown, and near-complete restabilization was observed upon ablation of both NAA10 and ZER1/ZYG11B (Fig. R1b).

Fig. R1 | a Stability analysis of SFLH-fused GFP with ZER1 and ZYG11B double knock out in NAA10 knockdown HEK293T cells by GPS. **b** Stability analysis of CFLH-fused GFP with ZER1 and ZYG11B double knock out in NAA10 knockdown HEK293T cells by GPS.

2. Evidence for full length substrates of ZER1/ZYG11B starting with Ala/Ser need to be provided. This can be done by an educational guess of known Nt Ala/Ser proteins that have a perfect ZER1/ZYG11B degron consensus and might be potential substrates. Alternatively, mass spec experiment of NAA10 silenced cells should recover endogenous substrates whose protein levels are decreased as a result of N-terminal acetylation inhibition.

Response:

We thank the reviewer for this suggestion. As you mentioned, we analyzed the interactors of ZER1 and ZYG11B using the BioGRID database (<https://thebiogrid.org/>) and found that the ACHAP (acetylcholinesterase-associated protein) bearing an Nt-Ser is a potential substrate. Expectedly, overexpression of ZER1 or ZYG11B caused a reduction in ACHAP levels (Fig. R2a-b). Furthermore, in contrast to WT ZER1, the mutant W552A or N579A, which would abolish the Nt-Ser binding, failed to induce the ACHAP degradation (Fig. R2c-d). These data have been added in the revised manuscript.

Fig. R2 | Western blotting analysis of ACHAP (acetylcholinesterase-associated protein) stability. **a** Stability analysis of ACHAP full-length protein with overexpression of ZER1 or ZYG11B in HEK293T cells by western blotting. **b** The relative protein levels of ACHAP were quantified with Gel Image System (Tanon-5200) and normalized to α -tubulin according to the results of three experiments. Error bars represent the standard error of mean (S.E.M.). * $p < 0.05$ (Student's t-test). **c** Stability analysis of ACHAP full-length protein with overexpression of WT and indicated mutant ZER1 proteins in HEK293T cells by western blotting. **d** The relative protein levels of ACHAP were quantified with Gel Image System (Tanon-5200) and normalized to α -tubulin according to the results of three experiments. Error bars represent the standard error of mean (S.E.M.). * $p < 0.05$ (Student's t-test); N.S., no significant differences.

Minor comment:

Experiments might work better in siRNA (and not shRNA) NAA10 depleted cells. The knockdown achieved with shRNA is very good (Fig. S2D), however the acetylation remained high and hardly changed (Fig. S2C), which might suggest

compensatory mechanism activated in NAA10 KD cells.

Response:

Thank you for the suggestion. As you suggested, we tried to use siRNA to suppress the expression of NAA10. Quantitative reverse transcription PCR (RT-qPCR) validation showed that all three different NAA10-siRNA sequences dramatically decreased the mRNA levels of NAA10 (Fig. R3a). However, the protein levels of NAA10 were not knocked down by the NAA10-siRNA (Fig. R3b), probably due to the long half-time of NAA10 in cells. Instead, the shRNA targeting NAA10 markedly reduced the protein levels (Fig. R3c), so we chose shRNA to carry out this experiment.

Fig. R3 | **a** Quantitative real-time PCR analysis to detect the relative mRNA levels of NAA10. HEK293T cells were treated with siRNA (control, 1, 2, 3) for 24 h after transient transfection. Error bars represent the standard error of mean (S.E.M.). * $p < 0.05$; ** $p < 0.01$; *** $p < 0.001$ (Student's t-test). **b** Western blot analysis of the relative protein levels of NAA10. HEK293T cells were treated with siRNA (control, 1, 2, 3) for 24 h after transient transfection. **c** Western blot analysis to evaluate the knockdown efficiency of NAA10 at protein levels in HEK293T cells infected with PLKO.1-shNAA10 lentivirus for 4 days.

Response to Reviewer #2

Major comments:

In the manuscript "CRL2/ZER1 recognizes small N-terminal residues for degradation" by Li et al., the authors have examined the specificity of the Cullin2 RING E3 ligase (CLR2) substrate receptors (namely ZER1 and ZYG11B).

The authors used a range of biochemical and structural approaches to demonstrate that both ZER1 and ZYG11B have an extended specificity, beyond the original defined specificity (i.e. recognition of N-terminal (Nt) Gly). Here they have focused largely on Nt-Ser and Nt-Ala, although they also clearly demonstrate that Nt-Thr and Nt-Cys

are also recognised by ZER1, similarly Nt-Cys is also recognised by ZYG11B. Although the recognition of Nt-Cys (by ZER1 and ZYG11B) was not initially defined using the peptide array (because of the limitation of this method), the identification of Nt-Cys as a primary destabilising residue (primary degron) in vitro, is very interesting. To date, Nt-Cys has only been identified as a secondary destabilising residue (following oxidation) and attachment of a primary destabilising residue. Therefore, I believe that the identification of Nt-Cys as a primary destabilising residue should be further examined/validated in cells, using the GPS assays, in normal and NatA deficient cells.

Response:

We thank the reviewer for the great suggestion. It is known that Cys-starting N-termini exist as both Nt-acetylated and unacetylated states (about 50%) in eukaryotic cells. To test whether ZER1 and ZYG11B can directly target Nt-Cys for degradation in cells, we first overexpressed ZER1 or ZYG11B in the Nt-Cys reporter cell lines. The GPS assays showed that the Cys-GFP fusion proteins are efficiently degraded upon ZER1 or ZYG11B overexpression (Fig. R4a). Moreover, the Cys-GFP exhibits dramatic instability in response to NAA10 KD (Fig. R4b), whereas concurrent ablation of ZER1/ZYG11B is sufficient to restore Cys-GFP stability (Fig. R4b), indicating that the free Nt-Cys can be recognized by ZER1/ZYG11B and target them for degradation.

To further substantiate the important role of ARM (armadillo) domain in recognizing Nt-Cys in cells, we mutated the W552 or N579 to alanine in the Flag-tagged full-length ZER1. The GPS assays showed that unlike WT ZER1, overexpression of W552A or N579A mutant did not efficiently target the Cys-GFP fusion proteins for degradation in the cells (Fig. R4c). Taken together, we demonstrate that Nt-Cys can be recognized as a primary destabilising residue by ZER1/ZYG11B for degradation. These data have been added in the revised manuscript.

Fig. R4 | **a** Stability analysis of CFLH-fused GFP upon ZER1 or ZYG11B overexpression in HEK293T cells by GPS. The ratio of GFP/RFP was analyzed by flow cytometry. **b** Stability analysis of CFLH-fused GFP upon NAA10 knockdown or with simultaneous knockout of ZER1/ZYG11B in HEK293T cells. **c** Stability analysis of CFLH-fused GFP with overexpression of WT and mutant ZER1 proteins in HEK293T cells.

Minor comments

1. Regarding the title, the authors only mention ZER1, even though the manuscript examines both CLR2 receptors (ZER1 and ZYG11B) equally. The authors should consider a change to the title to reflect the full extent of their data.

Response:

Many thanks for the great suggestion. We have changed our manuscript title into “CRL2^{ZER1/ZYG11B} recognizes small N-terminal residues for degradation”.

2. There is a misuse of articles (i.e. the) throughout the manuscript e.g. “degradation in the NatA-deficient cells”, should not include “the”. “Quality control through Gly/N-degron pathway”, should include “the” before Gly/N-degron. Likewise, “Through Ac/N-degron pathway”, should include “the” before Ac/N-degron.

Response:

Thank you for pointing out these mistakes and we have corrected them in the whole manuscript.

3. There are several references throughout the manuscript describing the binding of ZER1/ZYG11B to non-Ac/N-degrons, the authors need to be more precise with the terminology as ZER1/ZYG11B does not bind to all non-Ac/N-degrons, only a selection of non-Ac/N-degrons.

Response:

According to your kind suggestion, we have modified the terminology throughout the revised manuscript to make it more precise. Some related references of “non-Ac/N-degrons” are contextually closely connected, thus avoiding misunderstanding of this statement.

4. Line 48, as the modification occurs to the N-terminus, and is not facilitated by the N-terminus the following sentence on line 48 “The resulting Nt-Ac changes the chemical properties of the N-terminus by neutralizing its positive charge, and generally prevents the N-terminus from other modifications.” should be revised to something like “prevents other modifications of the N-terminus”

Response:

According to your kind suggestion, we have revised this sentence as the reviewer suggested.

5. At several places in the manuscript the authors appear to avoid discussing the recognition of Nt-Cys (and Nt-Thr). E.g. Line 99 – the authors identified that ZER1 binds Nt-Ser, Nt-Thr and Nt-Ala in preference to Nt-Gly, however they do not mention Nt-Thr here – even though it is clearly shown (and discussed) later in the manuscript. Again on Line 116 – the authors focus only on Nt-Ser and Nt-Ala, although both Nt-Thr and Nt-Cys (in the case of ZER1) exhibit stronger binding than Nt-Gly.

The authors should adjust the text to include all their data.

Response:

Thank you for your suggestions, we have added all the relevant data and adjust that in our revised manuscript.

6. Line 209 D556A should read D556

Response:

We are sorry for this typo, we have corrected this in the revised manuscript.

7. The discussion contains a few imprecise/incomplete statements/definitions that should be revised. The Arg/N-degron pathway (via UBR1) is not only responsible for the recognition of type 1 residues Nt-Arg, -Lys and -His (via the UBR box) but also

responsible for the recognition of type 2 residues (Nt-Trp, -Phe, -Tyr, -Leu and -Ile) as well as proteins bearing an Nt-Met-hy sequence (via the ClpS domain site within UBR1) (Xia et al., 2008; Schuenemann et al., 2009; Kim et al., 2013).

In addition to the recognition of Nt-Pro, -Ile, -Leu and -Phe, GID4 is able to recognise Nt-Val.

Response:

Thank you for your carefulness for our manuscript, we have revised the related statements according to the reviewer's comments in the revised manuscript.

8. Line 237 the following statement “the N-recognin is likely to target variable Nt-residues” should be revised, as the N-recognin does not target variable residues, rather it targets a defined selection of specific Nt-residues.

Response:

We agree with the reviewer and have revised this statement accordingly in the revised manuscript.

REVIEWERS' COMMENTS

Reviewer #1 (Remarks to the Author):

The authors largely addressed the reviewers comments on their manuscript.
As such the manuscript is improved and no more experiments are requested from my side.

Reviewer #2 (Remarks to the Author):

The authors have performed new experiments to address my previous concerns (raised during the initial review of the manuscript). These new data (and associated changes to the text) have been added to the revised manuscript. Overall, I am satisfied with the authors responses and their changes to the revised manuscript.

REVIEWERS' COMMENTS

Reviewer #1 (Remarks to the Author):

The authors largely addressed the reviewers comments on their manuscript. As such the manuscript is improved and no more experiments are requested from my side.

Response:

We greatly appreciate the reviewer's time and effort to review our manuscript.

Reviewer #2 (Remarks to the Author):

The authors have performed new experiments to address my previous concerns (raised during the initial review of the manuscript). These new data (and associated changes to the text) have been added to the revised manuscript. Overall, I am satisfied with the authors responses and their changes to the revised manuscript.

Response:

We greatly appreciate the reviewer's time and effort to review our manuscript.